# Application of a Monte Carlo Procedure for Probabilistic Fatigue Design of Floating Offshore Wind Turbines

Kolja Müller[1], Po Wen Cheng[1]

[1]Stuttgart Wind Energy @ Institute of Aircraft Design, University of Stuttgart, Germany

*Correspondence to*: Kolja Müller (mueller@ifb.uni-stuttgart.de)

**Abstract.** Fatigue load assessment of floating offshore wind turbines poses new challenges on the feasibility of numerical procedures. Due to the increased sensitivity of the considered system with respect to the environmental conditions from wind and ocean, the application of common procedures used for fixed-bottom structures results in either inaccurate simulation results or hard-to-quantify conservatism in the system design. Monte Carlo based sampling procedures provide a more realistic

approach to deal with the large variation of the environmental conditions, although basic randomization has shown slow convergence. Specialized sampling methods allow efficient coverage of the complete design space, resulting in faster convergence and hence a reduced number of required simulations. In this study, a quasi-random sampling approach based on Sobol' sequences is applied to select representative events for the determination of the lifetime damage. This is calculated applying Monte Carlo integration, using subsets of a resulting total of 16200 coupled time-domain simulations performed with

the simulation code FAST. The considered system is the DTU 10MW reference turbine installed on the LIFES50+ OO-Star Wind Floater Semi 10MW floating platform. Statistical properties of the considered environmental parameters (i.e. wind speed, wave height and wave period) are determined based on the measurement data from Gulf of Maine, USA. Convergence analyses show that it is sufficient to perform around 200 simulations in order to reach less than 10% uncertainty of lifetime fatigue damage equivalent loading. Complementary in-depth investigation is performed focusing on the load sensitivity and the impact of outliers (i.e. values far away from the mean). Recommendations for the implementation of the proposed methodology in the

design process are also provided.

## 1 Introduction

The site specific design (or site specific design verification) of floating offshore wind turbines (FOWT) requires the structure to withstand both the ultimate and fatigue limit states (ULS, FLS). While ULS loads represent worst case scenarios that can

be described by discrete combinations of extreme environmental conditions, the fatigue evaluation is more complex. This is due to the required consideration of two major tasks:

1. **Damage assessment through time domain simulations**

Due to the simultaneous occurrence of wind, wave and current loads as well as the complex structural interactions of the components within the system (i.e. RNA, tower, substructure and station keeping system) and the fact that the system behaviour is changing with the wind speed due to controller actions means that simplified methods (i.e. uncoupling of turbine and substructure or frequency domain analysis) are not recommended for the fatigue damage assessment. Instead, fully coupled, time domain simulations are commonly performed for the fatigue evaluation in the certification process. For fatigue analysis on a system level, rainflow counting and the Palmgren-Miner assumption (Fatemi & Yang, 1998) are typically applied in a post-processing step to determine the cumulated damage on a considered component. Compared to bottom-fixed offshore wind turbines the simulation time is increased due to (1) added complexity of hydrodynamics (floating body and mooring system) and (2) increased simulation time (3x1hr rather than 6x10min simulations is common due to the increased impact of the wave environment).

2. **Consideration of a complete characterization of the environment**

For fatigue load assessment it is not sufficient to calculate the loads during extreme events only, as the nature of fatigue is the accumulation of the damage over time. This requires the consideration of all relevant load scenarios over the expected life time of the system and typically this means a thorough investigation of the environmental conditions of the considered site. Environmental conditions that may affect fatigue loads of principal FOWT components are:

- Wind direction, wind speed, turbulence intensity, wind shear
- Wave direction, wave height, wave period
- Wind-wave misalignment, yawed inflow
- Current direction, current speed
- Ice, marine growth, etc.

It is clear that certain environmental conditions will have a larger impact on the fatigue loading. However, it is difficult to know this information a-priori if no sensitivity study is performed for the considered structure. This can be done in several ways which consider non-monotonous impact of independent parameters such as decision trees, neural networks, chi-square tests, regression analyses, variance based analyses or extended Fourier amplitude tests, as used in (Kusiak & Zhang, 2010), (Faerron Guzmán, Müller, & Cheng, 2018), (Müller, Reiber, & Cheng, 2016), (Hübler, Gebhardt, & Rolfes, 2017), (McKay, Carriveau, Ting, & Johrendt, 2014).

Compared to bottom-fixed offshore wind turbines the sensitivity of the system with respect to environmental conditions is increased, meaning that a larger amount of environmental conditions needs to be considered with a sufficiently high resolution. In particular the importance of wave period and directionality is increased.

With these requirements the basic problem for fatigue evaluation of floating wind turbines can be summarized with the so-called curse of dimensionality: Even though existing tools, based on engineering models, are capable of calculating the coupled structural loads of FOWT within reasonable simulation time, the sheer number of simulations that need to be carried out in order to consider all possible events makes it unfeasible to follow such a brute-force approach. As an example, a resolution of 2 m/s for wind speeds is typically required in design guidelines, leading to around 10 simulations for this dimension. If we consider this to be a reasonable resolution for each environmental dimension, the number of simulations to be carried out would be, following the formula $n_{sim} = 10^{n_{envCond}}$, 1000 simulations by only considering the most relevant parameters the wind speed, the wave height and the wave period. This problem already exists for the fixed-bottom offshore turbines and can lead to conservative estimates of the environment, i.e. the so called lumping of load cases (Kühn, 2001) or the use of damage equivalent wave heights (Passon, Damage equivalent wind-wave correlations on basis of damage contour lines for the fatigue design of offshore wind turbines, 2015), (Passon & Branner, Condensation of long-term wave climates for the fatigue design of hydrodynamically sensitive offshore wind turbine support structures, 2016), (Krieger, et al., 2015), which have found their ways into common guidelines (Det Norske Veritas AS, 2014). The approach is also considered as a state-of-the-art for floating wind turbines (Ramachandran, Vita, Krieger, & Müller, 2017). Simplified approaches impose a reduced environmental model to the design (typically wave periods are considered a function of wave heights, which are in turn depending on the wind speed) and hence may not be able to represent the observed spread of loads of the real turbine (Müller, Reiber, & Cheng, Comparison of Measured and Simulated Structural Loads of an Offshore Wind Turbine at Alpha Ventus, 2016). In particular for floating wind turbines, as described above, the longer simulation times and the increased number of simulations increase the importance of simplified approaches. However, due to the increased dimensionality of the problem, the use of simplified assumptions may lead to excessive conservatism or the overlooking of important impacts of the environment on the structure. For a more accurate, fully probabilistic design an approach is required which considers the load uncertainties imposed by the varying environmental conditions. In (Müller & Cheng, 2016) we indicated that a simple lumping of load cases as done in standard procedures may miss important load variation which is observed on the real turbine. In a follow-up presented in (Müller, Reiber, & Cheng, Comparison of Measured and Simulated Structural Loads of an Offshore Wind Turbine at Alpha Ventus, 2016) we showed that the spreading of full-scale environmental fatigue loads can be reproduced by state-of-the-art simulation tools when using appropriate methods for the selection of the environmental combinations. For the consideration of variations in environmental conditions, two general approaches are possible:

1. Determine a **representative** set of environmental conditions which represents the actual probabilistic environment with sufficient accuracy. The resulting representative set of loads can be used directly for the damage evaluation. The Monte Carlo method applied in the form of probability-based sampling best summarizes this approach. The initial idea of Monte Carlo integration is to replace the continuous average by discrete approximations of the average (Robinson & Atcitty, 1999).

2. Establish a **surrogate model** based on a predetermined set of environmental conditions, which describes the load behaviour of the considered system for all relevant environmental conditions. From the surrogate model, a Monte Carlo set resembling the occurrence probability can be determined efficiently for the damage integration. The main task of this approach is to determine a response function through regression analysis. A challenge is the determination of relevant points for the regression.

Some experience with both approaches has already been established in the past: (Graf, Stewart, Lackner, Katherine, & Veers, 2016) performed an in-depth comparison between simple Monte Carlo sampling and grid based analysis. (Choe, Byon, & Chen, 2015) investigated new sampling methods for reliability analysis with black box stochastic simulation tools (e.g. FAST) of wind turbine loads. (Stewart G. M., 2016) has looked into simplification methods for two different floating wind concepts, investigating bin reduction methods, probability-based sampling, surrogate models, and genetic programming. Further work on surrogate models was performed in (Zwick & Muskulus, 2016) and (Müller, Dazer, & Cheng, 2017), where simulations based on Latin Hypercube samples (LHS) were used as input for neural network models. Guanche et al. (Guanche, Guanche, Camus, Mendez, & Medina, 2013) used a maximum dissimilarity algorithm for design point selection and radial basis functions to determine a surrogate.

While the problem of dimensionality in accurate fatigue calculations has been addressed in different ways, a feasible universal procedure has not been presented up to this point (i.e. robust, efficient and accurate for major floating wind turbine components). In this way, suggested sampling procedures typically lead to unacceptable large simulation effort (especially with increasing dimensionality) and proposed surrogate models are either lacking accuracy or are tailored for a particular condition. The commonly used binning/gridding approach of environmental parameters either leads to a large simulation effort or overly conservative design (which still requires substantial insight into the sensitivity of component loads towards different environmental conditions that is obtained or documented by further simulation studies).

To test a universal and feasible solution for accurate fatigue calculation via Monte Carlo integration, this paper focusses on the determination of representative sets via sampling techniques and in particular on so-called quasi-random sampling. The general workflow is summarized in Figure 1. As described above, the bottleneck for the applicability of this procedure is the large number of dimensions in the environmental condition that need to be considered. In simulation applications, this is typically mitigated by using stratified (like LHS) or quasi-random sampling.

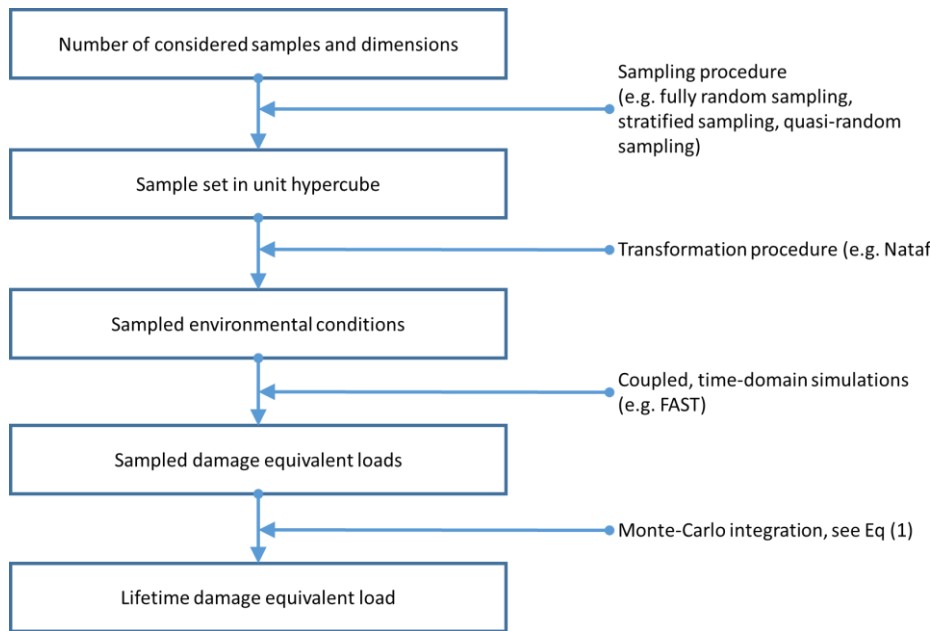

**Figure 1: workflow for FOWT damage assessment via sampling techniques**

## 1.2 Monte Carlo Integration

Fatigue damage assessment may be formulated as an integration problem, if the lifetime damage is defined as the accumulated (or integrated) damage over all load events that occur in the systems' lifetime. A continuous integration of the damage then requires an analytical function of the system response, which may be determined by regression analysis. In this work, a
5    numerical integration over the design space is performed by using the approximation of the continuous integration as defined in the Monte Carlo method (Metropolis & Ulam, 1949). The Monte Carlo integration replaces the integral by averaging the results of discrete evaluation points:

$$\int_{[0,1]^s} f(x)dx \approx \frac{1}{N} \sum_{i=1}^{N} f(\xi_i) \quad , \tag{1}$$

10   where $\xi_i \in [0,1]^s$ are each of the $N$ random, independent samples in the $s$-dimensional unit hypercube. A transformation with varying statistical distributions and ranges in each dimension is possible, which makes the method applicable for FOWT fatigue calculations.

Two approaches exist to describe the convergence and error behavior for the Monte Carlo integration.

The first one is based on the variance of the integrand $f(\xi_i)$. Through the central limit theorem, the root mean square error of the Monte Carlo method can be shown to be

$$\sigma_{MC} \to \frac{\sigma(f)}{\sqrt{N}} \text{ for } N \to \infty \quad , \tag{2}$$

Which shows the convergence rate of the Monte Carlo approach of $N^{-\frac{1}{2}}$. The relation of the error variance to the smoothness of the integrands has led to the variance reduction techniques such as importance sampling, LHS, stratified sampling, etc. (Singhee & Rutenbar, 2010), (Owen, 2013).

In a second approach, it can be shown that the root mean square error of the Monte Carlo method behaves according to the Koksma-Halwka inequality (Wang, 2001):

$$\int_{[0,1]^s} f(x)dx - \frac{1}{N}\sum_{i=1}^{N} f(\xi_i) \leq V(f)D_{N,s}^* \quad , \tag{3}$$

Where $V(f)$ is the Hardy and Krause variation of $f(x)$ (i.e. the integral of the absolute value of the gradient of $f$) and $D_n^*$ is the star-discrepancy of the random samples. The star-discrepancy is a measure of "*the uniformity of distribution of a finite point set*" (Wang, 2001).

The useable implication from Eq. (3) is that point sets with small star-discrepancy are preferable. Based on the goal of finding low-discrepancy point sets, infinite sequences in s-dimensional hypercube space have been constructed in the past. These lead to integration error bounds of order $O(N^{-1}\log^s N)$, and are weakly dependent on the dimensionality (Wang, 2001).

### 1.2.1 Quasi-Random Sampling

In this work, we will investigate the use of so-called "quasi-random" sampling techniques for the determination of a representative set of environmental conditions. The name "quasi-random" results from the fact that the design points are taken from sequences based on deterministic rather than random algorithms (primary motivation is to reduce discrepancy rather than variance of the integrand). They have shown better performance compared to LHS (Singhee & Rutenbar, 2010) and allow adding samples one-at-a-time (Romero, Burkardt, Gunzburger, Peterson, & Krishnamurthy, 2003), which is useful for convergence studies. The quasi-random sampling procedures covered here are the Sobol' sequences (Sobol, 1967), which can be described as low-discrepancy sequences, discrepancy being a measure of the uniformity of distribution of points across the unit hypercube. The algorithm used is based on the MATLAB implementation based on (Bratley & Fox, 2003). Figure 2 shows exemplary points in a unit square with increasing number of samples based on the sampling implementation used in this work. Other sequences have been defined, such as Halton, Faure and Niederreiter sequences, some of which perform better than Sobol' and may also be interesting in future applications. The interested reader is referred to (Niederreiter, 1992) for a more detailed description of the definition of quasi-random sequences.

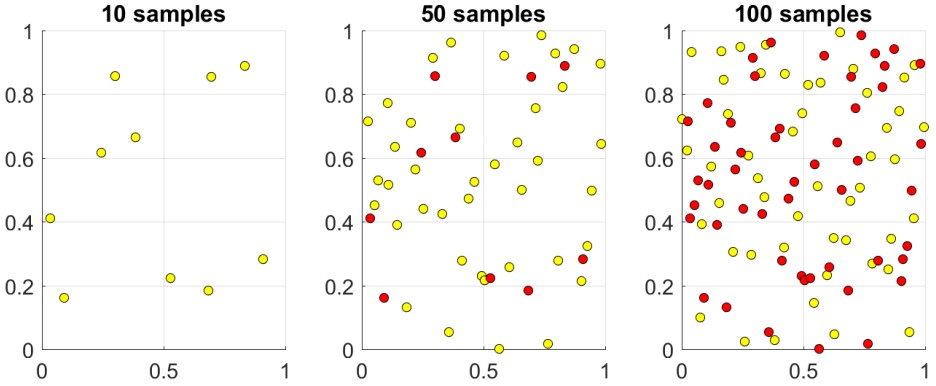

**Figure 2: exemplary presentation of samples based on Sobol' sequences with sample sizes 10, 50 and 1000. Red dots highlight points from the previous sample set.**

### 1.3 Considered System

The FOWT system considered in this study is the LIFES50+ public model of the OlavOlsen floater as presented in (Lemmer, Müller, Yu, Faerron-Guzman, & Kretschmer, 2016), see Figure 3. It is designed for the DTU10MW reference turbine (Bak, et al., 2013), which is positioned on a redesigned tower. As part of LIFES50+ efforts, platform and tower were designed for the medium severity site Gulf of Maine which is presented in (Krieger, et al., 2015) and (Ramachandran, Vita, Krieger, & Müller, 2017) and described in detail below.

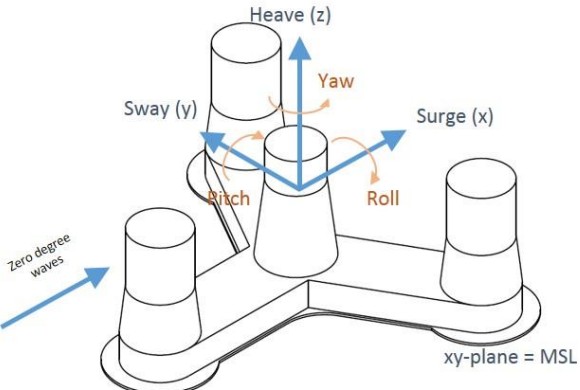

**Figure 3: sketch of the LIFES50+ public model of the OlavOlsen floater as used in this study.**

Simulations are carried out with FAST8v12 by NREL, using Blade-Element-Momentum theory for aerodynamic forces and first-order potential-flow theory as well as Morison drag forces for hydrodynamics. The linear potential-flow problem was solved for the equilibrium position (with draft of 22m) prior to the present work using the panel-code ANSYS AQWA (ANSYS

AQWA, 2018), resulting in the linear frequency-dependent hydrodynamic coefficients. Mooring line forces are determined dynamically using NRELs lumped mass mooring line modeler MoorDyn. Regarding the environmental conditions, the Kaimal spectrum and for the wave input the Jonswap spectrum was applied.

## 1.4 Considered Environment

Environmental parameters are based on Site B (medium severe site, reference site: Gulf of Maine, USA) as provided by the design basis of LIFES50+ (Krieger, et al., 2015). Three environmental conditions are considered in this study: wind speed, wave height and wave period. Wind and wave data are taken from measurement data from the NOAA buoy data network as was presented by (Stewart, Robertson, Jonkman, & Lackner, 2016). Hub height wind speeds are calculated using the power law (Gomez, Sanchez, Llana, & Gonzales, 2015) for the wind shear. Wind and wave directions are assumed co-aligned. Turbulence intensity is set according to IEC class C (International Electrotechnical Commission, 2005). Hourly measurements were evaluated between 2003 and 2015, resulting in an overall database of 103,282 useable measurement points.

## 2 Simulation Setup

### 2.1 Definition of sampling points

When using probabilistic samples, it is necessary to use a probability model of the considered environment. In this work, the samples are determined applying the idea of the Nataf transformation (Der Kiureghian & Liu, 1986), describing the environment by the linear correlation and marginal distributions of the considered variables. While this model may not be suitable for all sites, it is considered as adequate here for determining the performance of the presented procedure.

In a first step, $n$ sample points in three dimensional space are defined based on the Sobol' sequence as described above and stored in the Matrix $\boldsymbol{M} \in \mathbb{R}^{n \times 3}$. This provides data in the unit cube as shown in Figure 4, left. In the next step, the quasi-random data is transformed into standard Gaussian space by using the inverse CDF of a standard Gaussian distribution with zero mean $\mu = 0$ and unit standard deviation $\sigma = 1$ (Figure 4, center)

$$N = \phi^{-1}(\mathbf{M}) \in \mathbb{R}^{n \times 3} \quad . \tag{4}$$

The new sample matrix $\boldsymbol{N}$ has the empirical correlation matrix $\boldsymbol{R} \in \mathbb{R}^{3 \times 3}$. Subsequently, the environmental data is evaluated for the three considered environmental conditions wind speed, wave height and wave period. For this purpose, the correlation matrix $\boldsymbol{\rho} \in \mathbb{R}^{3 \times 3}$ is determined as well as the marginal distributions for each of the parameters. Afterwards, the Cholesky factorization can be applied for $\boldsymbol{\rho}$ and $\boldsymbol{R}$ to map the correlation of the measurement data to the sampled data:

$$O = B^{-1}AN \in \mathbb{R}^{n \times 3} \quad , \tag{5}$$

Where $A \in \mathbb{R}^{n \times 3}$ and $B \in \mathbb{R}^{n \times 3}$ are the upper triangle matrices resulting from the Cholesky factorization of the correlation matrices $\rho$ and $R$, respectively. In the final step, the marginal distributions of the measurement data are applied to obtain the final sampling data points $x'_i \in \mathbb{R}^{n \times 1}$ by using the cumulative distribution function for each of the parameters.

$$x'_i = F_i^{-1}(\phi(x_i)) \quad , \tag{6}$$

where $x_i$ are the sampled values from $O$, and $F_i^{-1}$ is the inverse marginal cumulative distribution function of the considered dimension (Figure 4, right).

Figure 5 shows the resulting sampling data compared to the baseline measurements and Figure 6 indicates the distribution of samples across the considered environmental parameters. It is noted that some error is introduced in this step due to the introduction of an environmental model (e.g. unrealistically small periods for large wave heights). For the current work, the environmental model is assumed accurate, although other models may be more accurate (see, e.g. (Graf, Stewart, Lackner, Katherine, & Veers, 2016)). Future studies need to determine the impact of errors introduced due to environmental modeling.

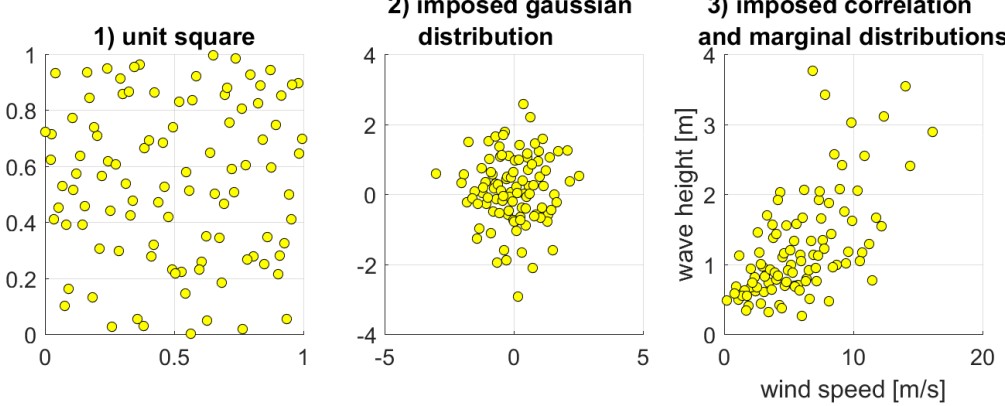

Figure 4: exemplary definition of environmental sample points based on 100 samples

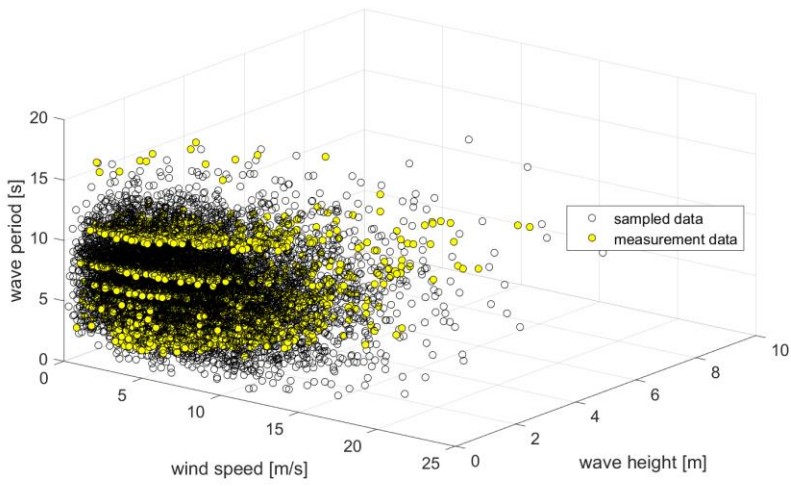

**Figure 5: sampled environmental conditions and baseline measurements**

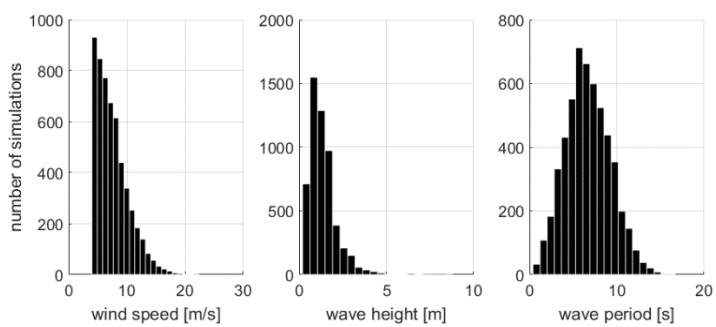

**Figure 6: histograms indicating marginal distributions of sampled environmental conditions**

### 2.2 Simulation settings

In this study a total of 5400 environmental points are used, which are considered to be sufficient for the convergence of the
5   results (see, e.g. (Graf, Stewart, Lackner, Katherine, & Veers, 2016), (Müller, Dazer, & Cheng, 2017)). For each environmental
point, three different wind and wave seeds are applied, resulting in a total of 16200 simulations. The total simulation length
considered for each environmental point is set to 3 hours (3 seeds of 1 hour simulation time each) as required by the LIFES50+
design basis (Ramachandran, Vita, Krieger, & Müller, 2017). Each seed is composed of a random wave field of 1hr length and
a periodic wind seed of 10 minute length which is repeated 6 times to obtain a 1 hour long wind field. Using repeated 10min
10   wind fields based on 10min environmental measurements leads to slightly conservative results, as the 10min measured mean

wind speed is assumed to be equal to the hourly mean (however, the hourly mean should be lower than 10min mean). A run-in time of 600s is added to each simulation in order to mitigate influences of transients at the beginning of the simulations. In addition, wind speed dependent initial conditions for the simulations (e.g. rotor speed), are determined previously using still water conditions. The considered run-in time is based on a previous investigation of the time series, the added use of initial

conditions as well as common values used in literature, see e.g. (Haid, et al., 2013). Note that some (possibly significant) uncertainty is added to the obtained load response by using only a limited number of seeds. The resulting uncertainty from using a limited number of wind and wave seeds is investigated in (Müller, Faerron Guzmán, Manjock, Vita, & Nipper, 2018). Generally, any uncertainty can be compensated by considering a higher percentile (e.g. 75[th]) of the considered seeds in order to obtain conservative results. In this exemplary study, simply the mean value of the results from the different seeds is used

for the analysis. Also, compared to state-of-the-art simulation, a much higher resolution of both wind speed (0.1m/s) and wave height (0.1m) is possible and implemented, hence the simulation uncertainty consideration is generally improved.

**2.3 Post Processing: Damage Equivalent Loads**

For the considered locations (blade root flap-wise bending moment, tower base fore-aft bending moment, leading fairlead tension), rainflow counting was applied to obtain the distribution of the load amplitudes $\Delta L$ for each time series and the

Palmgren-Miner linear damage accumulation law (PM) was used to calculate damage equivalent load amplitudes $\Delta L_{DE}$ (commonly known as damage equivalent loads or DEL) of the obtained 1 hour time series:

$$\Delta L_{DE,Simulation} = \left( \sum \frac{\Delta L_i^m \cdot n_i}{N_{ref,Simulation}} \right)^{\frac{1}{m}} \quad , \tag{7}$$

where $\Delta L_i$ are the load amplitudes of the time series and processed by rainflow counting, $n_i$ are the number of occurrences of the detected load cycles, $N_{ref,Simulation}$ is the reference cycle number applied for each simulation (set to $2 \times 10^6$ in this study) and $m$ is the slope of the SN-curve. In this study, $m = 4$ was assumed for all evaluated positions. This may not be adequate

for all positions (in particular for composite materials typically $m = 10$ is used (Det Norske Veritas AS, 2013)), but is regarded as sufficient for the demonstration purpose of the method.

After obtaining DELs for all the time series, the mean of common seeds was determined in order to obtain representative DELs for each combination of environmental parameters. Finally a lifetime equivalent DEL may be calculated based on all the considered samples:

$$\Delta L_{DE,Lifetime} = \left( \frac{N_{ref,Lifetime}}{N_{ref,Simulation}} \sum_{i=1}^{n_{samples}} \Delta L_{DE,i}^m \right)^{\frac{1}{m}} \quad , \tag{8}$$

where $N_{ref,Lifetime}$ is the reference cycle number for the full life time of the system. $N_{ref,Lifetime}$ is calculated by weighting each simulation according to their relative occurrence probability over the entire life time: $N_{ref,Lifetime} = w_{sim} \cdot N_{ref,Simulation}$. When performing a Monte Carlo type evaluation, the weighting factor is given by the ratio between simulation and overall life time, i.e. $w_{sim} = \frac{1}{n_{samples}} \cdot \frac{t_{life}}{t_{simulation}}$, with $t_{life} = 20 years$ in this study.

For more detailed DEL analysis, we may introduce the sum of all contributing DELs,

$$S_{DEL} = \sum_{i=1}^{n_{samples}} \Delta L_{DE,i}^m \quad , \tag{9}$$

which may be regarded as the variable term of the DEL definition in this work. This value is later used for normalization to investigate the impact of single large DELs on the lifetime DEL.

## 3 Results

In this study, the evaluated positions are the single simulation and lifetime DELs (subscripts DE and DELT) for the **blade root**
**flap-wise bending moment** ($\Delta M_{DE,BRF}$ and $\Delta M_{DELT,BRF}$) given in kNm, the **tower base fore-aft bending moment** ($\Delta M_{DE,TBFA}$ and $\Delta M_{DELT,TBFA}$) given in kNm and the **fairlead tension of the leading mooring line** ($\Delta F_{DE,FL1T}$ and $\Delta F_{DELT,FL1T}$) given in N. The positions are considered to provide a good overview of the loading across the overall system, with representative loads for the main system components, i.e. the rotor-nacelle-assembly, the tower and the mooring line system. Figure 7 shows a histogram of the resulting DELs from the simulation study for the different load positions. It can be observed
that for both tower base as well as the mooring lines, converged statistics have been obtained (Figure 7, center and right, shape of histogram not expected to change by adding additional simulations). For the blade root loads, the controller has a significant impact with respect to the resulting distribution of DELs which can be observed through the isolated, large peaks in the histogram (Figure 7, left). All three positions show distributions of the loading with multiple peaks. The origin of this is discussed further below as well.

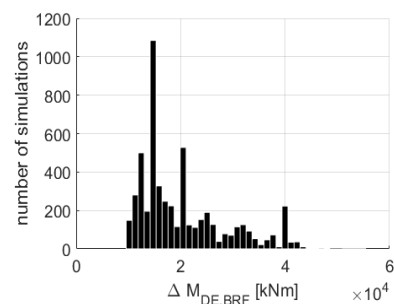
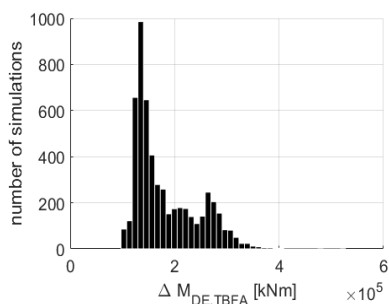
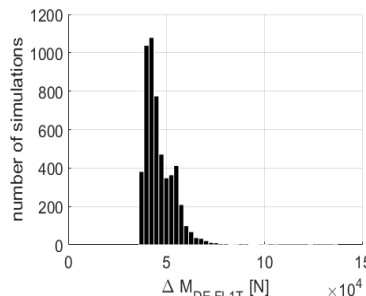

**Figure 7: histograms of simulated DELs for different positions**

Using the resulting sets of DELs, the life time equivalent DEL can be calculated as described above. In order to evaluate the convergence of the applied process, the number of considered samples $n_{samples}$ has been varied. The Sobol' sequence allows us to consider only a subset of the complete set of simulations while maintaining the good space-filling properties of the sampling procedure. For this purpose, the number of considered samples is reduced and the resulting sub-sequence of considered samples is shifted along the original Sobol' sequence. The resulting data is processed through a quasi-random bootstrap analysis based on all possible combinations available for each number of considered simulations (resulting in 4920-5400 samples). The bootstrap analysis provides statistical information of the uncertainty of the lifetime damage for different numbers of samples, which is presented in Figure 8.

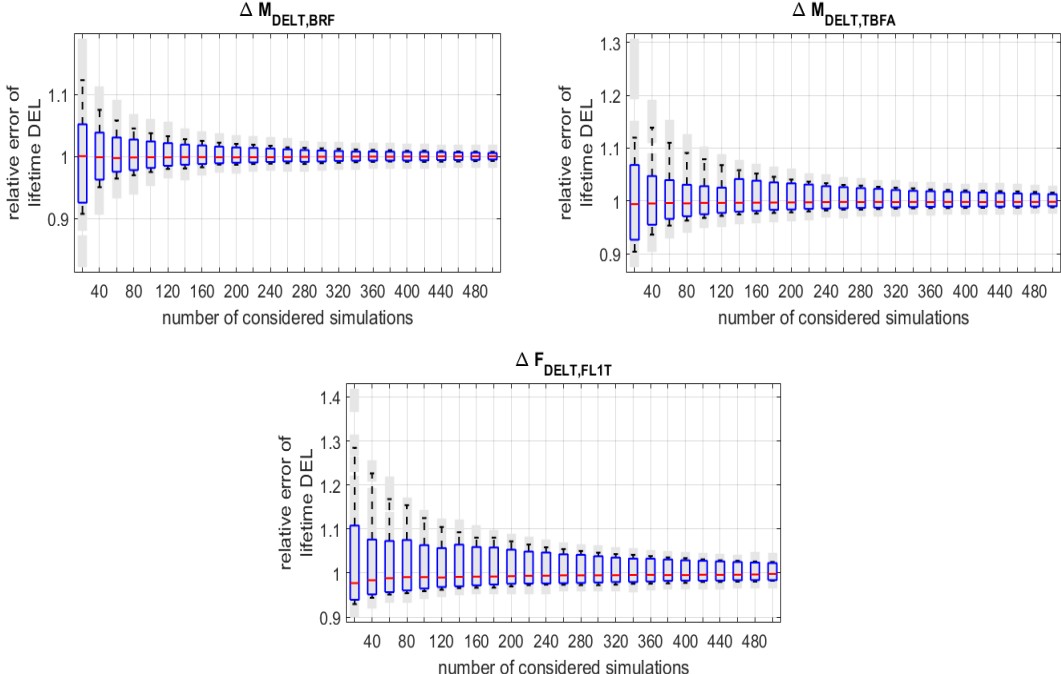

**Figure 8: Bootstrap analysis of resulting lifetime DEL with increasing number of considered samples/simulations. Results are normalized with the resulting value based on 5400 samples. Boxplots are indicating 1st /99th percentile values (whiskers), 10th /90th percentiles (box frames) and median values (red lines). Grey boxes indicate results from individual evaluations (baseline data for statistical evaluation).**

The results show a fast convergence of the lifetime DEL calculation, with about 140 samples it is possible to obtain a 99% confidence limit within the 10% error margin. As mentioned further below, for the damage assessment, to be below 10% error a total number of 200-500 samples are necessary, depending on the considered component. The results also show that an underestimation of the damage is slightly more likely than an overestimation when considering a small number of samples (median below 1 for small number of considered simulations), which may result from the impact of large DELs and could be

of interest to investigate in a follow-up study. However, a significant overestimation (> 10% error) is much more likely than a significant underestimation (1st percentile greater than 0.9 for all considered positions and number of considered samples), which makes the procedure conservative overall.

The number of required samples for fatigue load assessment is roughly one order higher than common approaches for fatigue assessment, which would require a total of 48 samples (considering wind speeds from 4-20m/s with 2m/s resolution, one representative wave height per wind speed and 3 representative wave periods per wave height). This is however a minimum effort estimate which also does not include consideration of a separate sensitivity analysis, which is required with this approach. Looking from the application perspective, a safety factor would be required due to the possible underestimation of the lifetime DEL. If the goal of the evaluation is to provide a sufficiently conservative estimate of the lifetime DEL, 20 samples with a safety factor of 1.1 seems to be a reasonable choice from the presented results. However, this could lead to very conservative results, with estimated lifetime DELs up to 30% larger than the actual one. Thus, increasing the number of considered samples reduces the required safety factor as well as the conservatism in the design.

Figure 9 shows the summary 1st and 99th load percentiles for the different load locations, indicating both the required safety factors (based on the 1st percentile) as well as the maximum possible overestimation (99th percentile) as a function of considered samples.

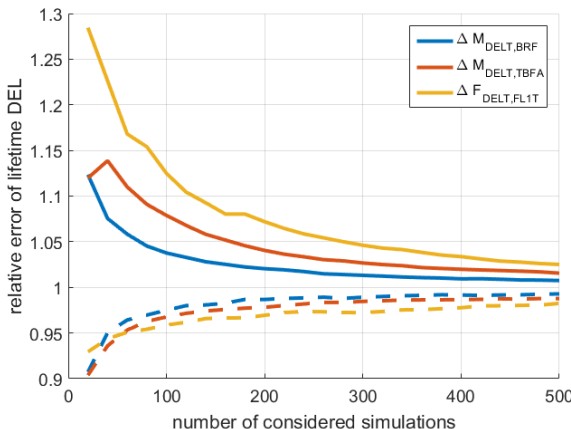

**Figure 9: 1st percentile (dashed lines) and 99th percentile (straight lines) of the computed lifetime DEL for different load locations as a function of considered samples/simulations. Results are normalized with the resulting value based on 5400 samples.**

A question that arises is the origin for the uncertainty when only a small number of samples are considered. This is quite different for the different locations (i.e. from around 20% range for *RootMyb1* up to almost 40% for *FAIRTEN1*) and is closely linked to the most extreme DEL values that were simulated (similar to large load values observed in measurements). The reduced impact of these *extreme* values with increasing number of simulations is further addressed in the discussion section below.

## 4 Discussion

This section will address two topics which were highlighted before:

1) Influence of environmental parameters on the DEL distribution
2) Influence of large simulated DELs on the lifetime damage value

### 4.1 Impact of environmental parameters

As mentioned before, the distributions of the sampled DELs in Figure 7 indicate multiple peaks. Typically, multiple peaks in distributions may originate from overlapping single-peak distributions. Figure 10 shows histograms of the different positions after binning the data into three distinctive wind speed regions. It is visible how different wind speed domains result in a different single-peak load distribution across the whole system. Fatigue assessment needs to be able to take this varying load behavior into account. Well distributed samples are able to capture this, while it is hard to predict analytically.

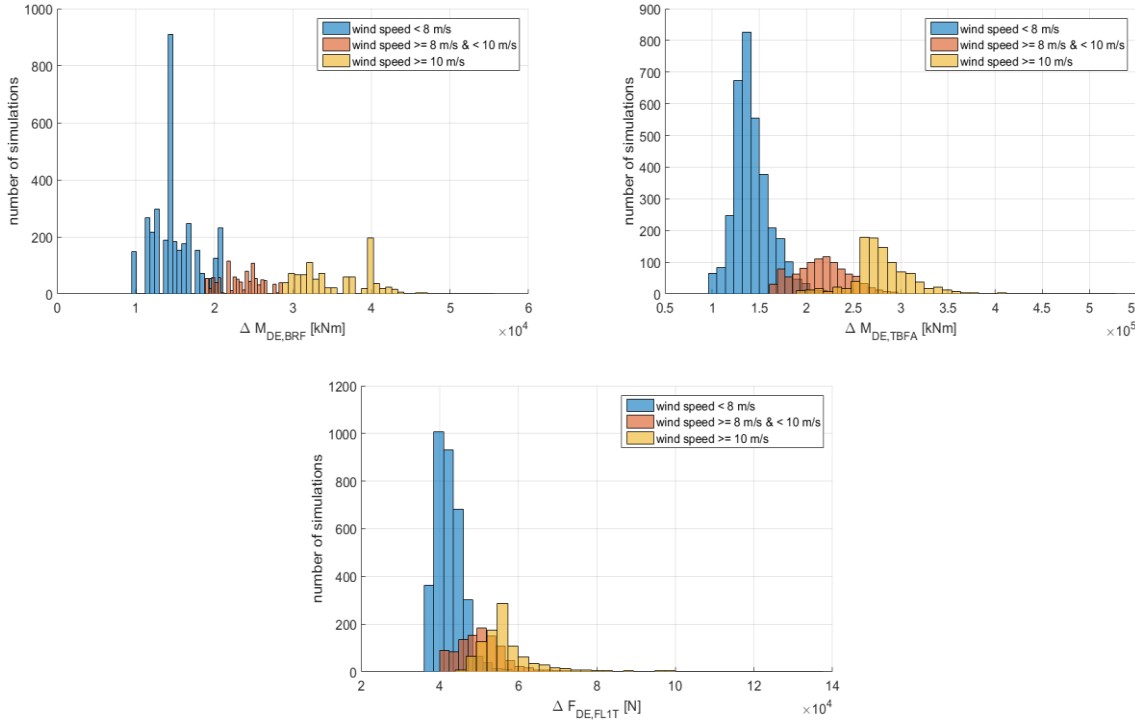

**Figure 10: histograms of simulated DELs across different wind speed ranges for different load positions.**

Additionally, the influence on the DEL from different environmental conditions can be investigated through scatter plots as presented in Figure 11. It is clearly visible how the influence of the wind speed is reduced with decreasing height of the observed position (decreasing slope, Figure 11, left column). On the contrary, the influence of the wave height is increasing with decreasing height of the observed position (increasing slope, Figure 11, center column). No general trend of impact of the wave period can be observed for any of the observed positions.

However, distinctive DEL peaks are present at position-specific wave periods. This has been investigated in a related study, which was based on the same system and environment, but focused in particular on the sensitivity of DELs towards different environmental conditions. Figure 12 shows the results of this study regarding sensitivity on wave height and period. The plots show that for large wave heights, distinctive wave periods can lead to maximum fatigue loads. A closer look on the

hydrodynamics in Figure 13 reveals the close connection of this effect to the wave excitation magnitude. The peak in the wave excitation magnitude is around the same period as observed from the simulation results at the tower base (i.e. 0.65 rad/s or 9.67s). However, this period is not exactly equal for the tower base and blades (around 7-8s, see Figure 12) and in particular not for the observed mooring line (around 5-6s, see Figure 12). This indicates that the increased model fidelity in coupled time-domain simulations can lead to a different period for maximum loads than the period derived from the panel code frequency

domain results suggest, which are based only on the floating platform (neglecting the RNA, tower and mooring lines). This effect is in particular of interest for the state-of-the-art fatigue design approach, where particular periods need to be selected which are expected to lead to increased loading in order to reach conservative results. The results from this study show that there are periods, which can lead to increased fatigue loading. These periods are linked to, but not equal to, peaks from the hydrodynamic wave excitation of the platform. It was observed that an analysis with unit-waves of changing periods may

provide a correct indication of where the periods leading to possible maximum loads are. They are position-dependent and are expected to be – as the hydrodynamic wave excitation – direction dependent as well. Even though this effect does not have a significant influence on the loads shown in this study, the importance of this effect may increase at more severe sites, with larger waves. Fatigue analysis based on well distributed samples may be able to consider this effect by definition, due to the high resolution for each environmental condition. Grid based procedures are only able to capture this effect if the grid is given

an adequate resolution, which significantly increases the simulation effort. Simplified grids based on occurrence probability of certain wave periods (e.g. considering 3 wave periods for each wave height only) are likely to ignore this effect, because they do not take into consideration the component sensitivity towards distinctive wave periods.

**Blade root flapwise bending moment**

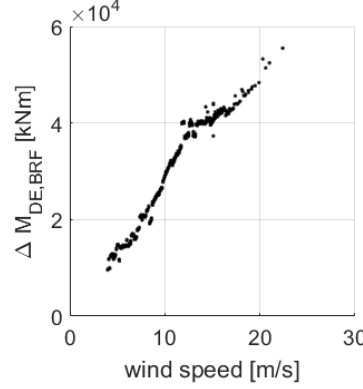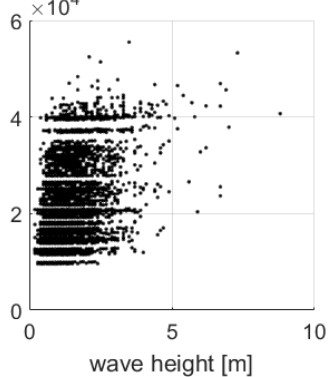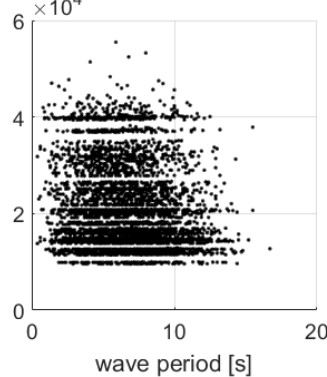

**Tower base fore-aft bending moment**

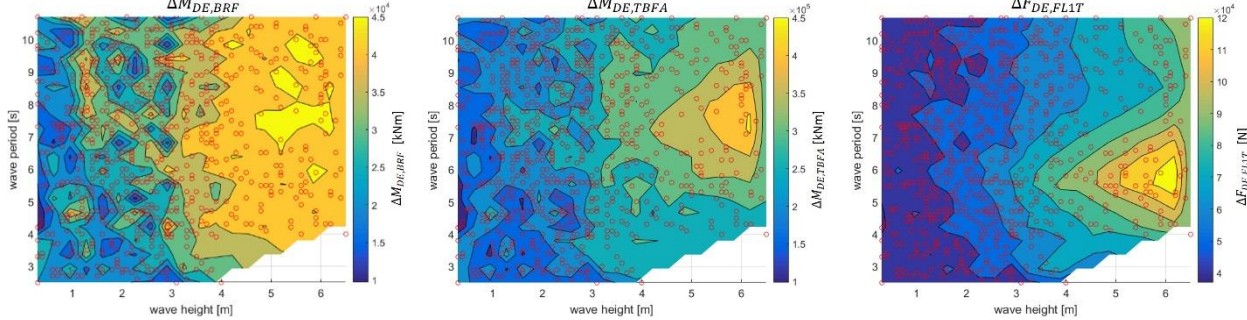

Figure 11: scatterplots of simulation results showing influence of different environmental positions on the DEL at different positions.

Figure 12: DEL contour plots for blade root flap-wise bending moment, tower base fore-aft bending moment and fairlead 1 tension. Plotted for all load ranges and showing channel-specific excitation periods. Red dots indicate performed simulations. Results are obtained independent of wind speed.

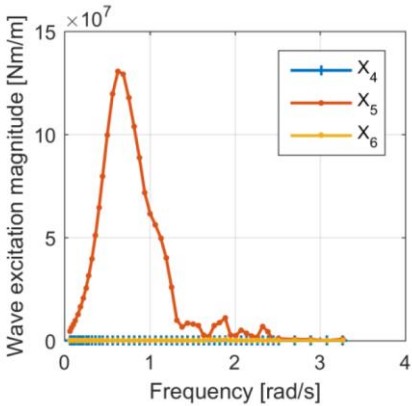

**Figure 13: wave excitation magnitudes for roll ($X_4$), pitch ($X_5$) and yaw ($X_6$) degree of freedom for the considered platform and considered direction (Yu, et al., 2017).**

## 4.2 Damage contribution from high DEL events (impact of outliers)

This section addresses the large damage contribution of large DEL events. These are rare combinations of environmental variables which result in large DELs and thus effectively may have an increased impact on the fatigue loading than more likely events which appear more often but have a much smaller damage contribution. The term outlier is chosen here to highlight the

very small probability of occurrence rather than indicating an error (which sometimes is used in analysis of measurement data). This is closely linked to the Palmgren-Miner rule as summarized in Equation (7). In it, the DEL from a single event is elevated to the exponent of the SN-curve slope $m$ before being included in the overall sum of single DELs. This leads to a higher-weighting of larger DELs and lower-weighting of events with small DEL contribution. This effect is investigated more closely here, as this can affect the applicability of the presented methodology (due to the possibility of 'missing' important (damaging)

events through the chosen sampling procedure). It is highlighted that based on the results presented above, there is no evidence that such 'outliers' can have a significant effect when sufficiently large sample sizes are considered and this is elaborated further here below.

Following the definition for DELs in Equation (7), the impact of single events may be determined through normalizing the single DELs with the overall sum of all the events, i.e. by defining a normalized DEL-contribution variable:

$$C_{DE,i}^* = \frac{\Delta L_{DE,i}^m}{S_{DEL}} \quad . \tag{10}$$

Note that the sum of all normalized DEL-contributions $\sum_i C_{DE,i}^*$ is equal to 1. $C_{DE}^*$ is useful to give an indication of the relative contribution of a single DEL to the sum of all DELs, which was defined as the variable part of the DEL definition in equation (9). This simplification is not taking into account the summation rules for DEL values.

Additionally, we may define two DELs $\Delta L_{DE,1}$, $\Delta L_{DE,2} = \mathrm{f}(\Delta L_{DE,i})$, which are composed of a number of DELs $\Delta L_{DE,i}$. Suppose that $\Delta L_{DE,1}$ is considering an additional DEL with a large magnitude $\Delta L_{DE,1} = f(\Delta L_{DE,i}, \Delta L_{DE,large})$. Then, the ratio between the DELs is defined as:

$$r_{\Delta L_{DE}12} = \frac{\Delta L_{DE,1}}{\Delta L_{DE,2}} = \left(1 + \frac{\Delta L_{DE,large}^m}{S_{DEL}}\right)^{\frac{1}{m}} \quad . \tag{11}$$

Note the similar expression of the second summand on the right hand side compared to the definition of a normalized DEL in
equation (10). Equation (11) lets us evaluate the impact of a single, large DEL on the lifetime DEL.

Based on these definitions, it is possible to analyze the impact of large, singular DELs on the lifetime DEL with an increasing number of considered samples/simulations. For this purpose, the largest singular DEL from the results was taken and added to the DEL-sets of varying sample size. Hence, the effect of considering an increasing number of simulations could be evaluated. Figure 14 shows the accumulation of normalized DEL contributions as defined in equation (10) towards the full sum of
contributions as described in equation (9) (Figure 14, left). In this plot, the contributions are sorted by size (starting with the smallest contribution) and the accumulated DEL-contribution is plotted for increasing indices. This way, the contribution of the final (largest) DEL becomes obvious: It can be seen most clearly for $n_{sim} = 10$, where adding the final DEL leads to more than quadrupling of the accumulated DEL (increase by 80% when increasing the index from 9 to 10 or normalized index from 0.9 to 1.0). The relative contribution of the largest DEL varies with the considered set of chosen DELs. The statistics of this,
dependent on the number of considered samples are shown in the next plot (Figure 14, center).

Now, the determined possible contribution to the sum of DELs has to be evaluated taking into account the complete definition of DEL as defined in equation (8). This can be done by assuming a relative increase of the sum of DELs as presented in Figure 14, center, and evaluating the relative increase of lifetime DELs according to equation (11). This is shown in an inversed way below (Figure 14, right), in order to allow a visual transfer of the results from the central plot to the right plot. It can be seen
that even though a single, large DEL may lead to a change of 20% of the total sum of DELs, the effect on the lifetime DEL is only around 5%. Based on the results from Figure 14, with only 50 simulations it is sufficient to reach a maximum impact of the largest DEL of 10% (i.e. 10% relative increase of DEL => ca. 45% increase of normalized DEL=> ca. 50 simulations; see dotted lines in Figure 14), which is well within the uncertainty limits for simulation studies on fatigue analysis. In order to mitigate any possible errors due to 'missing' significant severe events, an additional safety factor may be applied. The
decreasing impact of large DELs with increasing number of considered simulations explains the fast conversion of the lifetime DEL obtained through the sampling procedure in this study. It also indicates a quite robust behavior of damage towards extreme (i.e. rare and severe) environmental conditions if a sufficient amount of samples is considered.

On a final note, it needs to be added that in this study, a focus was put on investigating DELs and not the resulting damage, which might be more interesting from an industrial point of view. Due to the characteristics of the SN-curve to be used it can

be shown that, considering the same component and equal number of reference cycles, any increase of the DEL has an effect on the final damage according to

$$\frac{D_1}{D_2} = \left(\frac{\Delta L_{DE,1}}{\Delta L_{DE,2}}\right)^m \qquad . \tag{12}$$

Exemplary, a DEL increase of 5% then leads to an increase of damage of about 21.6%. Hence, if looking at damage contribution the results of this study have to be adjusted accordingly and lead to a slower convergence. In (Müller & Cheng, Validation of uncertainty in IEC damage calculations based on measurements from alpha ventus, 2016), we obtained a damage uncertainty (range of 95[th] percentiles for different wind speeds) for fixed bottom offshore fatigue simulations according to (International Electrotechnical Commission, 2009) between 10 to 20%. To reach the same uncertainty in this study, 200-500 samples are required (see Figure 14, center).

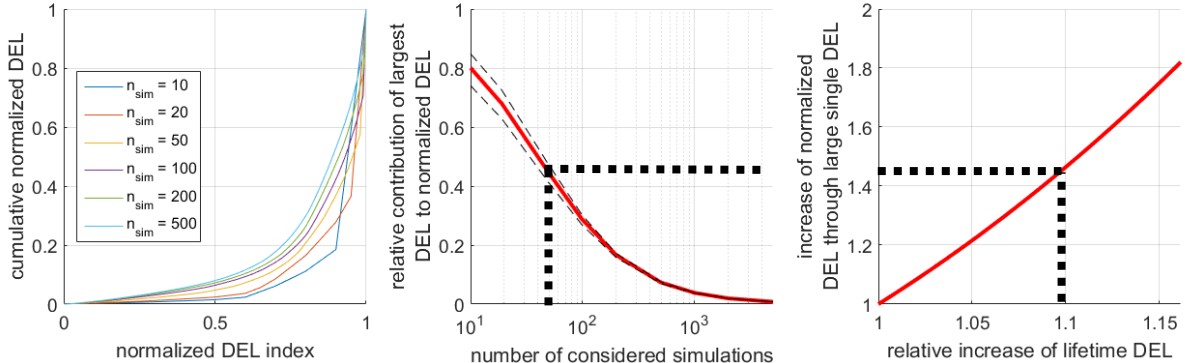

**Figure 14: cumulative normalized DEL-contribution ($C^*_{DE,i}$) for different numbers of considered simulations, exemplary results for one sample (left), contribution of the largest DEL to the normalized DEL with increasing number of considered simulations, showing median, 10[th] and 90[th] percentiles (center) and impact of increase of the normalized DEL on lifetime DEL (right). Plots show results for tower base fore-aft bending moment. Dotted lines indicate connection between center and right plot. Normalized DEL index used in left plot to allow showing varying number of DELs.**

## 5 Summary, Conclusions and Outlook

The present study shows a probabilistic fatigue analysis of floating wind turbines based on Monte Carlo integration. Based on the assumptions used in this work regarding system setup and environmental modeling, the use of Sobol' sequences as a quasi-random sampling method provides satisfactory convergence of the lifetime damage estimate for the analyzed load positions located in the RNA, tower and mooring system. The applied sampling method inherently allows a convergence study through the definition of the chosen low-discrepancy sequence. This is a significant advantage over stratified sampling methods like the well-known LHS which require resampling for changing numbers of samples. The uncertainty in lifetime DEL reduces

rapidly, reaching a satisfactory level between 100 and 200 samples (or 200-500 samples when considering overall damage). The knowledge on the quantified uncertainty depending on the number of considered simulations may be used in the future for the definition of safety factors. This enables the designer the chance to decide on (1) a fast approach, using a small number of simulations with larger safety factors or (2) a detailed approach, using a large number of simulations with small safety factors. In-depth analysis of the results show the high potential for design sensitivity studies. Also the ability to consider varying importance of environmental parameters for different positions is given by the sampling procedure. The possibility to consider the full spectrum of all relevant environmental parameters is one major advantage of sampling procedures compared to state-of-the-art procedures. The state-of-the-art methods rely on arbitrary experience of the engineers with respect to the most severe loading conditions and can lead to inaccurate and inconsistent results. Results from sampling procedures may be treated similar to measurements, which can lead to a more realistic representation of the system behavior. A minimum number of simulations is required in order to reduce the potentially larger influence of extreme values that can lead to overestimate of life time damage. The required number of samples found in this study lies well within the limits of feasibility for fatigue load analysis and may be accompanied by further safety factors if seen fit. More experience is needed to achieve resilient safety factors for different floaters and components, number of simulations, number and type of environmental conditions.

Further work will focus on more extensive fatigue evaluations, taking into account more environmental parameters. Furthermore, the quality of the environmental model and the sampling procedure may be developed further. Sequential sampling may be interesting to further reduce the simulation effort without sacrificing the accuracy of the estimate. Finally, the procedure needs to be applied for different loading environments and FOWT systems in order to verify the general applicability of the procedure.

## 6 Acknowledgements

The present work is carried out as part of the LIFES50+ project, which has received funding from the European Union's Horizon 2020 research and innovation programme under grant agreement No 640741. The funding and support is gratefully acknowledged. Also, we are grateful to Dr. techn. Olav Olsen AS for the permission and contribution to set up the public semi-submersible design based on their concept of the OO Star Wind Floater (www.olavolsen.no).

## List of Abbreviations

| | |
|---|---|
| CDF | Cumulative Distribution Function |
| DEL | Damage Equivalen Load |
| DTU | Danmarks Tekniske Universitet |
| FLS | Fatigue Limit State |
| FOWT | Floating Offshore Wind Turbine |
| LHS | Latin Hypercube Sampling |
| NOAA | National Oceanic and Atmospheric Administration |

| | |
|---|---|
| NREL | National Renewable Energy Laboratory |
| RNA | Rotor Nacelle Assembly |
| ULS | Ultimate Limit State |

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
