# Peer review of "Application of a Monte Carlo Procedure for Probabilistic Fatigue Design of Floating Offshore Wind Turbines"

_Wind Energy Science, 2017_

## Referee Comment (RC1) · Anonymous Referee #1 · 27 Nov 2017

Dear authors, I would like to thank you for the interesting work on a potential new, more flexible, methodology, to perform the fatigue life assessment and verification of FOWT. I think that this material is original enough to be published, but I made a number of major and minor comments in order to enhance its readability and potential to be adopted by the larger researcher community.

Please also note the supplement to this comment:
https://www.wind-energ-sci-discuss.net/wes-2017-41/wes-2017-41-RC1-supplement.pdf

**Supplement:**

Dear authors,

I would like to thank you for the interesting work on a potential new, more flexible, methodology, to perform the fatigue life assessment and verification of FOWT.

I think that this material is original enough to be published, but I made a number of major and minor comments in order to enhance its readability and potential to be adopted by the larger researcher community.

**MAJOR**
- Page 1, Line 24: "The site specific design of floating offshore wind turbines (FOWT) requires…"

Having talked with some of the floating support structure designers and manufacturers, the tendency seem to be to be able to have wind turbine agnostic designs, as well as applicable to geographical areas rather than specific locations.

I think in here it would be better to say the "site specific verification of the design against fatigue…"

- Page 3, line 28: "Some experience with both approaches has already been established in the past:…"
It is not clear to the reviewer the improvement, or the potential improvement, of the proposed methodology with respect to the methodologies listed here.
There should be a clear paragraph/couple of statements here about this aspect.

- Page 5, line 2: "The potential-flow model was established prior to the present work using the panel-code Ansys AQWA"
This statement is quite general, and the details of the approach used in Ansys AQWA should be described.

- Page 10, line 2: "In this study a total of 5400 environmental points are used, which are considered to be sufficient for the convergence of the results."
Could you please justify this statement? This should be done by either showing any sensitivity analysis done, or citing relevant previous articles/reports.

- Page 10, line 7: "A run-in period of 600s is added to each simulation in order to mitigate influences of transients at the beginning of the simulations"
Has a sensitivity analysis been done to determine this value? Or is it based on previous experience? Please justify an/or cite relevant previous literature.

- Page 18, line 9: "Figure 14 shows the accumulation of normalized DELs from equation (10) towards the full sum of DELs as described in equation (9) (Figure 14, left)."
The explanation for the left graph of figure 14 should be expanded, since at the moment it is not very clear. Examples, similarly to the example given for the middle graph, should be given.

**MINOR**
- Page 2, line 5: "Due to the simultaneous occurrence of wind, wave and current loads as well as the complex structural interactions of the components within the system (i.e. RNA, tower, substructure and station keeping system)"
It seems an incomplete statement.

- Page 2, end, and Page 3, start, and throughout the article

"(Passon, Damage equivalent wind-wave correlations on basis of damage contour lines for the fatigue design of offshore wind turbines, 2015), (Passon & Branner, Condensation of long-term wave climates for the fatigue design of hydrodynamically sensitive offshore wind turbine support structures, 2016),"

No need to cite the title as well.

---

## Short Comment (SC1) · 1 Dec 2017

Dear authors,

Thank you for your contribution that covers a really interesting topic.

The paper addresses the important subject of fatigue load assessment for floating offshore wind turbines. It proposes a probabilistic, sampling based approach instead of the state-of-the-art bin based approaches. For this sampling based approach, an advanced quasi-random sampling is proposed that is supposed to lead to better results than plain Monte Carlo sampling.

Starting from set of more than 10000 time domain simulations with different environmental conditions, sampled from three correlated statistical distributions, subsets with smaller numbers of simulations are selected, and the resulting uncertainty is analysed. It is concluded that 100 to 200 samples are sufficient to reach satisfying accuracy.

Although the topic is really interesting and the paper is mostly well written, some modifications or clarifications are needed. And an assessment of the performance of the current approach should be added (see remark 23).

1) P.2, l.19: Ocean currents are not really relevant for fatigue, as they induce nearly no dynamic loads. Hence, leave it out, if you do not have a reference that proves the relevance of currents.
2) P.2, l.21: Although, there is no sensitivity analysis for the considered structure, studies for other structures could be a good starting point. For example, you chose your three environmental conditions, based on some knowledge ("most relevant parameters" p.2, l.30). Hence, it would be good to cite a sensitivity analysis e.g. [1].
3) P.2, l.29: You correctly state that a high number of simulations is needed due to the "curse of dimensionality". However, here, you do not say anything about the additional number of random seeds that are needed (later on, you use 3 seeds). However, the stochastic nature of wind and waves increases the number of simulations. This effect can be significant and should be mentioned here.
4) P.3, l.28-34: You say that there are two approaches (improved sampling and surrogate modelling), and "some experience with both approaches has already been established in the past" (p.3, l 28). However, the literature mentioned in the following focuses nearly completely on surrogate models, while your work uses improved sampling. There should be some references to work concerning improved sampling as well, even if the references might not cover exactly the problem of fatigue of floating wind turbines (e.g. [2])
5) P.7, l.1: Some more information on the simulation setup would be nice, e.g. turbulent wind model, wave spectrum etc.
6) P.10: Figure 6 (left) and Figure 5 seems to be not consistent by cutting the wind speed distribution at cut-in wind speed.
7) P.10, l.3: You are using "only" 3 seeds. This is not a lot, if you compare it, for example, with the results in [3, 4]. However, you can argue that you are, firstly, using 1h simulations, and secondly, the use of a probabilistic, sampling approaches inherently includes different seeds (for sampling points with similar parameters). However, this should be briefly discussed here. Otherwise, the reader could claim that 3 seeds are not sufficient.
8) P.11: Eq. 9: You introduce this quantity as "your" DEL. However, it might be really confusing for the reader that this is NOT a DEL, but an $DEL^m$. This also changes the unit to e.g. $(kNm)^4$. Either rename it, or redefine it and stick to "real" DELs. This is really important, as you switch between

real DELs and DEL$^m$ throughout the paper. This is confusing and also introduces the problem of different uncertainties in Section 4.2.

9) P.12: Figure 7: Units of the DELs are missing.

10) P.12, l.5: How many bootstrap samples do you use?

11) P. 13, l. 1: You state that you reach a fast convergence, as a 10% error margin is obtained by using 140 samples. However, later on you discuss correctly that this error is valid for DELs and not for damages or lifetimes. In the end, the designer is mainly interested in damages of lifetimes. Your 10% error margin leads to 40-50% errors in the damage for m=4, or even errors far above 100% for m=10. You briefly discuss this topic later on, but it has to be mentioned here, as it somehow relativises the results.

12) P.14: Figure 10: Are the units correct? If you are showing "your" DEL (Eq. 9), then you have other units. And add units to all subplots in Figure 10.

13) P.14, l.14: You state that there is no significant impact of the wave period. Later on, you discuss the effect of the wave period. This contradicts each other. Hence, reformulate the statement.

14) P.15, l.19: Sample based approaches consider this effect by definition. What about state-of-the-art bin based approaches? It should be possible to include these effects by introducing wave period bins. However, it is obvious that sampling based approaches are beneficial here.

15) P. 15: Figure 11: Check units of "your" DELs again.

16) P. 16: Figure 12: You cut off a region of high wave heights and low wave periods. This is correct. But still, it seems as if there are still quite high waves for small periods (e.g. 6.5m with 4s). These are quite special conditions that you normally do not see (e.g. FINO data [5], or standards [6] $11.1\sqrt{H_s/g} = 9.0s$). Are these conditions actually present at the considered site (in the raw data), or is it only due to the defined correlation?

17) P. 17, l.8: "over-weighting" sounds as if this weighting would be wrong. Perhaps "high" and "low weighting" is better.

18) P.17: Eq. 10: Again, this is not really a DEL, but DEL$^m$. Either rename it, or redefine it.

19) P.18, l.9: "Figure 14 shows the accumulation of normalized DELs from equation (10) towards the full sum of DELs as described in equation (9)". You should probably say that you are starting with the sample with the lowest damage. (see remark 21 as well)

20) P.18, l.18: You say that a large DEL (actually DEL$^m$) can change the sum of DELs (DEL$^m$) by 20%, but the lifetime DELs (real DEL) only by 5%. However, if we are going back to damages or lifetimes, we have again 20%. Hence, the 20% are the more important number (or the 45% in line 20, compared to 10%). This should be clarified.

21) P.19: Figure 14 (left): The figure is not directly self-explaining. I would add the highest DEL first (and not the lowest) so that you have a large increase for a small number of DEL indices. Furthermore, it might be helpful to rename the horizontal axis, e.g. "data proportion" or "sample proportion".

22) P.19, l. 15: 100-200 samples are only valid for these three environmental conditions. This should be mentioned here.

23) In general: A comparison of this approach with others is missing. If you want to demonstrate the performance of this approach, you should either compare it to the state-of-the-art binning method or to plain MCS. For example, you could add another figure, comparable to Figure 8, showing the convergence of MCS and of your low-discrepancy sequence approach for samples up to 200. Such a comparison would really make clear the advantages of your work.

Editorial remarks:

1) P.2, l.2: The sentence is incomplete.
2) P.5, l.9: n=N? Be consistent with the nomenclature
3) P.6, l.12: Space after "2017)"
4) P.7: The text in Figure 3 is barely readable. Improve quality of this figure.
5) P.10, l.15: Remove the second space after "from the".
6) P.10, l.16: Don't use the 2e6 notation.
7) P.12: Figure 7: Be consistent with your notation in the whole paper. Either use $1 \times 10^4$ or $1 \cdot 10^4$, not both.
8) P. 16: Figure 12: Please improve readability.

References

[1] Hübler, C., Gebhardt, C. G., & Rolfes, R. (2017). Hierarchical four-step global sensitivity analysis of offshore wind turbines based on aeroelastic time domain simulations. Renewable Energy, 111, 878-891.

[2] Choe, Y., Byon, E., & Chen, N. (2015). Importance Sampling for Reliability Evaluation With Stochastic Simulation Models. Technometrics, 57 (3), 351–361.

[3] Zwick, D., & Muskulus, M. (2015). The simulation error caused by input loading variability in offshore wind turbine structural analysis. Wind energy, 18(8), 1421-1432.

[4] Häfele, J., Hübler, C., Gebhardt, C. G., & Rolfes, R. (2017). A comprehensive fatigue load set reduction study for offshore wind turbines with jacket substructures. Renewable Energy, 118, 99-112.

[5] Scott, G. (2007). Applications and Analysis of Offshore Wind and Wave Measurements. In 52nd IEA Topical Expert Meeting Wind and Wave Measurements at Offshore Locations, Berlin. https://www.ieawind.org/task_11/TopicalExpert/52_Wind_Wave.pdf

[6] Guideline for the Certification of Offshore Wind Turbines. Germanischer Lloyd Rules and Guidelines, IV—Industrial Services, Part, 2005

---

## Referee Comment (RC2) · Anonymous Referee #2 · 17 Dec 2017

Paper WES-2017-41

This paper presents an interesting approach for fatigue design of floating offshore turbines. Paper is very well written and organised.  The following summarises some comments that for the authors' consideration before the paper is published in the journal:

Comments

1. Introduction – it is recommended to explain briefly in more detail why fatigue life estimation for FOWTs is more complicated than in the case of offshore wind turbines with rigid foundations.

2. References: No need to include the full title of the paper. Including author and year of publication should be enough.

3. All abbreviations (e.g. NATAF) need to be explained.

4. "Centre" not "center"

5. Page 2, Line 4: explain what simplified methods you are referring to.

6. Fig. 5: assess quantitatively the extent to which the sampled data plotted here represents statistically the 3D measurement data.

7. Page 2, line 7:  Add a reference to literature presenting the Palmgren-Miner fatigue life model.

8. Page 10, line 5: The explanation given in the following sentence is not sufficiently clear – "The total simulation length……using 6 wind fields of 10-minute length." Why did you use 6 wind fields? Did you use 10min average data to represent hourly conditions? I would expect that using 10 min data would be more conservative only when the same values are used to represent typical hourly values (averaging would reduce peaks).

9. Page 7, line 4 – add reference to literature of ANSYS AQWA©.

10. Page 10 – last para.: add references to literature recommending values of $m$ for different materials.

11. Page 13, 1$^{st}$ para: provide a physical reason for the underestimation at low number of samples.

12. Fig. 10 shows that the number of samples is small for the high wind speed regime (>10m/s). Higher wind speeds result in higher material stress. To what extent does the smaller number of samples yield reliable predictions for fatigue life at high wind speeds? Would constant speed operation for above rated windspeeds play a role here?

13. Comment on the computational cost savings when adopting the proposed approach when compared to existing methods.

14. It is recommended in the conclusions to suggest values for factor of safety to compensate for limitations of the method.

---

## Author Comment (AC1) · 31 Jan 2018

**Authors response**

- Page 1, Line 24: "The site specific design of floating offshore wind turbines (FOWT) requires…" Having talked with some of the floating support structure designers and Manufacturers, the tendency seem to be to be able to have wind turbine agnostic designs, as well as applicable to geographical areas rather than specific locations. I think in here it would be better to say the "site specific verification of the design against fatigue…"
Added "(or site specific design verification)". I do not want to replace site specific design as I believe it will be of interest again once a reliable production chain for FOWT is available (see bottom-fixed).

Page 3, line 28: "Some experience with both approaches has already been established in the past:…"
It is not clear to the reviewer the improvement, or the potential improvement, of the proposed methodology with respect to the methodologies listed here. There should be a clear paragraph/couple of statements here about this aspect.
**Added another paragraph describing improvements considered necessary for applicability.**

- Page 5, line 2: "The potential-flow model was established prior to the present work using the panel-code Ansys AQWA"
This statement is quite general, and the details of the approach used in Ansys AQWA should be described.
**Added some more information, hope this clarifies the performed analysis.**

- Page 10, line 2: "In this study a total of 5400 environmental points are used, which are considered to be sufficient for the convergence of the results."
Could you please justify this statement? This should be done by either showing any sensitivity analysis done, or citing relevant previous articles/reports.
**Added citations**

- Page 10, line 7: "A run-in period of 600s is added to each simulation in order to mitigate influences of transients at the beginning of the simulations"
Has a sensitivity analysis been done to determine this value? Or is it based on previous experience? Please justify and/or cite relevant previous literature.
**Added info and citations**

- Page 18, line 9: "Figure 14 shows the accumulation of normalized DELs from equation (10) towards the full sum of DELs as described in equation (9) (Figure 14, left)."
The explanation for the left graph of figure 14 should be expanded, since at the moment it is not very clear. Examples, similarly to the example given for the middle graph, should be given.
**Added more detailed explanation**

- Page 2, line 5: "Due to the simultaneous occurrence of wind, wave and current loads as well as the complex structural interactions of the components within the system (i.e. RNA, tower, substructure and station keeping system)"
It seems an incomplete statement.
**Fixed**

- Page 2, end, and Page 3, start, and throughout the article "(Passon, Damage equivalent wind-wave correlations on basis of damage contour lines for the fatigue design of offshore wind

turbines, 2015), (Passon & Branner, Condensation of long-term wave climates for the fatigue design of hydrodynamically sensitive offshore wind turbine support structures, 2016),"
No need to cite the title as well.

fixed

**wes-2017-41-RC2**

- Introduction – it is recommended to explain briefly in more detail why fatigue life estimation for FOWTs is more complicated than in the case of offshore wind turbines with rigid foundations

  Some paragraphs were included to highlight the added complexity and increased simulation effort compared to fixed bottom in the introduction.

- References: No need to include the full title of the paper. Including author and year of publication should be enough.

  fixed

- All abbreviations (e.g. NATAF) need to be explained.

  Added List of Abbreviations

- "Centre" not "center"

  This is a choice of the author to use American English rather than British English.

- Page 2, Line 4: explain what simplified methods you are referring to.

  added info: (i.e. uncoupling of turbine and substructure or frequency domain analysis)

- Fig. 5: assess quantitatively the extent to which the sampled data plotted here represents statistically the 3D measurement data.

  This is considered to be out of scope of the current work. We are developing tools in order to assess the uncertainty of joint environmental probability models. From my experience, this is extremely difficult for dimensions larger than one, but I am open for suggestions. I have added the following phrase in order to highlight the problem though: It is noted that some error is introduced in this step due to the introduction of an environmental model. For the current work, the environmental model is assumed accurate. Future studies need to determine the impact of errors introduced due to environmental modeling.

- Page 2, line 7: Add a reference to literature presenting the Palmgren-Miner fatigue life Model

  Added source

- Page 10, line 5: The explanation given in the following sentence is not sufficiently clear – "The total simulation length……using 6 wind fields of 10-minute length." Why did you use 6 wind fields? Did you use 10min average data to represent hourly conditions? I would expect that using 10 min data would be more conservative only when the same values are used to represent typical hourly values (averaging would reduce peaks).

  We used 10min periodic wind fields stitched together 6 times to obtain a one-hour wind field. The determination of the wind speeds was done based on 10min average values. By assuming the 10min average value to be simply repeated 6 times, we obtain an hourly wind field with the same average as the 10min value. For a one hour simulation however, this would result in overly conservative loading, as the hourly average wind speed with the same occurrence probability should be lower.

- Page 7, line 4 – add reference to literature of ANSYS AQWA©.

  added reference

- Page 10 – last para.: add references to literature recommending values of m for different Materials

  added reference

- Page 13, 1 st para: provide a physical reason for the underestimation at low number of samples.

  added: "…which may result from the impact of large DELs and could be of interest to investigate in a follow-up study"

- Fig. 10 shows that the number of samples is small for the high wind speed regime (>10m/s). Higher wind speeds result in higher material stress. To what extent does the smaller number of samples yield reliable predictions for fatigue life at high wind speeds? Would constant speed operation for above rated windspeeds play a role here?

  Fig. 11 shows that highest loads should be expected around rated for components located below the RNA (represented by flapwise blade root bending moment). Loads in the RNA are linked closely to the wind speed where a lower number of samples is unlikely to have an impact. The Impact and likelihood of overlooking unlikely events leading to large DELs is investigated in section 4.2. It was found to be within reasonable limits if the simulation number is sufficiently high. Constant speed operation (active blade pitch) above rated wind speed reduces the thrust force and the mean loads on lower-part components of the wind turbine.

- Comment on the computational cost savings when adopting the proposed approach when compared to existing methods.

  added the following: "The number of required samples for fatigue load assessment is roughly one order higher than common approaches for fatigue assessment, which would require a total of 48 samples (considering wind speeds from 4-20m/s with 2m/s resolution, one representative wave height per wind speed and 3 representative wave periods per wave height). This is without consideration of a separate sensitivity analysis, however, which is necessary in the common approach."

- It is recommended in the conclusions to suggest values for factor of safety to compensate for limitations of the method.

  As this is a first study with the method, we do not want to suggest any hard values at this point. They may be very dependent on the boundary conditions used in this study. We added the following: "More experience is needed to achieve resilient safety factors for different floaters and components, numbers of simulations and numbers and types of environmental conditions."

**wes-2017-41-SC1**

- P.2, l.19: Ocean currents are not really relevant for fatigue, as they induce nearly no dynamic loads. Hence, leave it out, if you do not have a reference that proves the relevance of currents.

  Agree that the currents are not expected to have a high contribution on dynamic loading. However, they may change the operating point of the system and in particular the mooring lines. This, in turn may lead to larger load amplitudes, especially in the mooring lines, which is why we like to keep it in the list.

- P.2, l.21: Although, there is no sensitivity analysis for the considered structure, studies for other structures could be a good starting point. For example, you chose your three environmental conditions, based on some knowledge ("most relevant parameters" p.2, l.30). Hence, it would be good to cite a sensitivity analysis e.g. [1]

  Thanks for the input, I added the following: "This can be done in several ways which consider non-monotonous impact of independent parameters such as decision trees, neural networks, chi-square tests, regression analysis, variance based analysis or extended Fourier amplitude test, as used in [sources]"

- P.2, l.29: You correctly state that a high number of simulations is needed due to the "curse of dimensionality". However, here, you do not say anything about the additional number of random seeds that are needed (later on, you use 3 seeds). However, the stochastic nature of wind and waves increases the number of simulations. This effect can be significant and should be mentioned here.

  Added the following statement:

  "Note that some (possibly significant) uncertainty is added to the obtained load response by using only a limited number of seeds. The resulting uncertainty from using a limited number of wind and wave seeds is investigated in [source]. Generally, any uncertainty can be compensated by considering a higher percentile (e.g. 75th) of the considered seeds in order to obtain conservative results. In this exemplary study, simply the mean value of the results from the different seeds is used for the analysis."

- P.3, l.28-34: You say that there are two approaches (improved sampling and surrogate modelling), and "some experience with both approaches has already been established in the past" (p.3, l 28). However, the literature mentioned in the following focuses nearly completely on surrogate models, while your work uses improved sampling. There should be some references to work concerning improved sampling as well, even if the references might not cover exactly the problem of fatigue of floating wind turbines (e.g. [2])

  Thanks again for the very interesting paper. I added this next to another paper I recently found on monte carlo sampling for floating wind turbines.

- P.7, l.1: Some more information on the simulation setup would be nice, e.g. turbulent wind model, wave spectrum etc.

  I added "Regarding the environmental conditions, the Kaimal spectrum and for the wave input the Jonswap spectrum was applied."

- P.10: Figure 6 (left) and Figure 5 seems to be not consistent by cutting the wind speed distribution at cut-in wind speed.

  We did not simulate any events below cut-in, hence they are not included in the considered environmental conditions.

- P.10, l.3: You are using "only" 3 seeds. This is not a lot, if you compare it, for example, with the results in [3, 4]. However, you can argue that you are, firstly, using 1h simulations, and secondly, the use of a probabilistic, sampling approaches inherently includes different seeds

(for sampling points with similar parameters). However, this should be briefly discussed here. Otherwise, the reader could claim that 3 seeds are not sufficient.

This was also addressed regarding an earlier comment. Using 3 seeds is according to the design basis developed in corporation with DNVGL and is thus seen as sufficient for the present study. We agree that further investigation is to be performed on the accuracy, however using sampling, a much higher number of wind seeds is used overall. Hence, I added the following statement: "Also, compared to state-of-the-art simulation, a much higher resolution of both wind speed (0.1m/s) and wave height (0.1m) is possible and implemented, hence the simulation uncertainty consideration is generally improved."

- P.11: Eq. 9: You introduce this quantity as "your" DEL. However, it might be really confusing for the reader that this is NOT a DEL, but an DEL m . This also changes the unit to e.g. (kNm) 4. Either rename it, or redefine it and stick to "real" DELs. This is really important, as you switch between real DELs and DEL m throughout the paper. This is confusing and also introduces the problem of different uncertainties in Section 4.2.

This was very helpful to get some feedback on these definitions. It is definitely not straightforward. I took another look at the topic and now changed quite a lot of items linked to this. I hope it is now clearer. The general idea is to look at the DEL values for this simplified approach as it is an easily digested and flexible value.

- P.12: Figure 7: Units of the DELs are missing.

Done

- P.12, l.5: How many bootstrap samples do you use?

The bootstrap is not the same for the Sobol sequences, as the order of indices needs to be maintained. Thus, the number of samples is varied slightly by increasing the number of considered simulations. Added the following info: The resulting data is processed through a quasi-random bootstrap analysis based on all possible combinations available for each number of considered simulations (resulting in 4920-5400 samples).

- P. 13, l. 1: You state that you reach a fast convergence, as a 10% error margin is obtained by using 140 samples. However, later on you discuss correctly that this error is valid for DELs and not for damages or lifetimes. In the end, the designer is mainly interested in damages of lifetimes. Your 10% error margin leads to 40-50% errors in the damage for m=4, or even errors far above 100% for m=10. You briefly discuss this topic later on, but it has to be mentioned here, as it somehow relativises the results.

I have updated the information regarding differentiation between damage and DEL throughout the document. I hope this helped to clear out any confusion.

- P.14: Figure 10: Are the units correct? If you are showing "your" DEL (Eq. 9), then you have other units. And add units to all subplots in Figure 10.

Plot is showing simulation DELs, no conversion is done. Note similarity between figures 7 and 10, they use the same data in a different way.

- P.14, l.14: You state that there is no significant impact of the wave period. Later on, you discuss the effect of the wave period. This contradicts each other. Hence, reformulate the statement.

Done

- P.15, l.19: Sample based approaches consider this effect by definition. What about state-of-the-art bin based approaches? It should be possible to include these effects by introducing wave period bins. However, it is obvious that sampling based approaches are beneficial here.

Added the following: "Grid based procedures are only able to capture this effect if the grid is given an adequate resolution, which significantly increases the simulation effort. Simplified grids based on occurrence probability of certain wave periods (e.g. considering 3 wave

periods for each wave height only) are likely to ignore this effect, because they do not take into consideration the component sensitivity towards distinctive wave periods."

- P. 15: Figure 11: Check units of "your" DELs again.

  Plot is showing simulation DELs, no conversion is done. Fairlead tension units are updated (kNm -> N)

- P. 16: Figure 12: You cut off a region of high wave heights and low wave periods. This is correct. But still, it seems as if there are still quite high waves for small periods (e.g. 6.5m with 4s). These are quite special conditions that you normally do not see (e.g. FINO data [5], or standards [6] $11.1\sqrt{H_s/g} = 9.0s$). Are these conditions actually present at the considered site (in the raw data), or is it only due to the defined correlation?

  This is due to the use of the Nataf model, which is a simple environmental model. It is not considered of relevance for this study, as we assume an accurate environmental model. However, it is clear that further study is needed in order to determine the error introduced through environmental models and the required follow-up actions. We highlighted this in the description of Figure 5.

- P. 17, l.8: "over-weighting" sounds as if this weighting would be wrong. Perhaps "high" and "low weighting" is better.

  Corrected.

- P.17: Eq. 10: Again, this is not really a DEL, but DEL m . Either rename it, or redefine it.

  I used a more clear definition now.

- P.18, l.9: "Figure 14 shows the accumulation of normalized DELs from equation (10) towards the full sum of DELs as described in equation (9)". You should probably say that you are starting with the sample with the lowest damage. (see remark 21 as well)

  Changed the wording for more clarity

- P.18, l.18: You say that a large DEL (actually DEL m ) can change the sum of DELs (DEL m ) by 20%, but the lifetime DELs (real DEL) only by 5%. However, if we are going back to damages or lifetimes, we have again 20%. Hence, the 20% are the more important number (or the 45% in line 20, compared to 10%). This should be clarified.

  From our point of view, it was more interesting to look at the DEL here as it is more flexible. We included notes throughout the document to highlight that convergence focusses on DEL and that the convergence for damage is higher.

- P.19: Figure 14 (left): The figure is not directly self-explaining. I would add the highest DEL first (and not the lowest) so that you have a large increase for a small number of DEL indices. Furthermore, it might be helpful to rename the horizontal axis, e.g. "data proportion" or "sample proportion".

  The way of sorting is seen as a personal preference. The x-axis actually resembles the normalized DEL index, i.e. [1-10]/10 or [1-500]/500. This is important to show the decreasing contribution of the final DEL value.

- P.19, l. 15: 100-200 samples are only valid for these three environmental conditions. This should be mentioned here.

  statement was added in the conclusion: More experience is needed to achieve resilient safety factors for different floaters and components, number of simulations, number and type of environmental conditions.

- In general: A comparison of this approach with others is missing. If you want to demonstrate the performance of this approach, you should either compare it to the state-of-the-art binning method or to plain MCS. For example, you could add another figure, comparable to Figure 8, showing the convergence of MCS and of your low-discrepancy sequence approach for samples up to 200. Such a comparison would really make clear the advantages of your work.

I agree that a direct comparison would be interesting. However, this would require a very large amount of extra simulation effort in order to be able to perform the bootstrap analysis. Also, state-of-the-art procedures aim at a conservative rather than an accurate design. It is well shown in literature that quasi-random sampling outperforms standard random sampling. Any theoretical study will show the same. We show here that with quasi-random sampling a feasible simulation effort can be achieved, which was the goal of this paper. I hope this clarifies why we do not choose to go into more depth on this.